# Separating the Signal from the Noise: How Psychiatric Diagnoses Can Help Discern Food Addiction from Dietary Restraint

**DOI:** 10.3390/nu12102937

**Published:** 2020-09-25

**Authors:** David Wiss, Timothy Brewerton

**Affiliations:** 1Department of Community Health Sciences, Fielding School of Public Health, University of California Los Angeles, Los Angeles, CA 90025, USA; 2Department of Psychiatry and Behavioral Sciences, Medical University of South Carolina, Charleston, SC 29425, USA; drtimothybrewerton@gmail.com

**Keywords:** food addiction, eating disorder, dietary restraint, substance use disorder, posttraumatic stress disorder, trauma, adverse childhood experience, early life adversity, psychiatric comorbidity, clinical vignette

## Abstract

Converging evidence from both animal and human studies have implicated hedonic eating as a driver of both binge eating and obesity. The construct of food addiction has been used to capture pathological eating across clinical and non-clinical populations. There is an ongoing debate regarding the value of a food addiction “diagnosis” among those with eating disorders such as anorexia nervosa binge/purge-type, bulimia nervosa, and binge eating disorder. Much of the food addiction research in eating disorder populations has failed to account for dietary restraint, which can increase addiction-like eating behaviors and may even lead to false positives. Some have argued that the concept of food addiction does more harm than good by encouraging restrictive approaches to eating. Others have shown that a better understanding of the food addiction model can reduce stigma associated with obesity. What is lacking in the literature is a description of a more comprehensive approach to the assessment of food addiction. This should include consideration of dietary restraint, and the presence of symptoms of other psychiatric disorders (substance use, posttraumatic stress, depressive, anxiety, attention deficit hyperactivity) to guide treatments including nutrition interventions. The purpose of this review is to help clinicians identify the symptoms of food addiction (true positives, or “the signal”) from the more classic eating pathology (true negatives, or “restraint”) that can potentially elevate food addiction scores (false positives, or “the noise”). Three clinical vignettes are presented, designed to aid with the assessment process, case conceptualization, and treatment strategies. The review summarizes logical steps that clinicians can take to contextualize elevated food addiction scores, even when the use of validated research instruments is not practical.

## 1. Background

The Yale Food Addiction Scale (YFAS) was created in 2009 to match criteria for Substance Abuse in the Diagnostic and Statistical Manual of Mental Disorders (DSM-IV) and has been validated as a tool for identifying eating patterns which resemble alcohol and drug addictions [1]. The YFAS 2.0 (released 2016) reflects updated criteria in the DSM-5 [2]. Prevalence estimates of food addiction (FA) in a nationally representative US sample are approximately 15%, with higher rates in those who are obese [3]. A meta-analysis of 51 studies suggests the mean prevalence of FA worldwide is 16.2% [4]. Unlike obesity, rates of FA in the US are elevated among individuals with higher incomes [3]. In other studies of patients with obesity seeking weight loss, prevalence estimates range from 6.7–16.5% [5,6] which closely mirror national prevalence rates for alcohol and substance use disorders (SUDs) [7]. Estimates are lower in adolescents [8], suggesting that FA develops over time. There has been a growing interest in early life psychosocial risk factors (e.g., trauma—defined as a deeply distressing or disturbing experience) in the development of both FA and obesity [9,10,11,12]. The current review employs a biopsychosocial perspective on FA, considering predisposing factors such as early life adversity which occurs in the first 18 years of life and can become biologically embedded, impacting reward function and eating behavior over the life course.

There has been considerable debate regarding the utility of an FA “diagnosis” without considering the contribution of dietary restraint in increasing FA symptoms [13,14,15,16,17]. While FA is not recognized by the DSM, the term diagnosis is used loosely throughout this manuscript. A common criticism of the YFAS in clinical applications is that the measure itself does not detect restrained eating (tendency to restrict food intake for weight control). The Disordered Eating and Food Addiction Nutrition Guide (DEFANG) attempted to conceptualize a role for FA into the common eating disorder (ED) paradigm which rejects the concept of food having addictive qualities by favoring an “all foods fit” (“no bad foods”) approach [18]. However, the DEFANG did not incorporate restrained eating into the framework, nor did it consider the impact of trauma. The current review aims to help clinicians consider divergent nutritional strategies in patients with elevated YFAS scores, and to avoid misconceptions regarding treatment. Importantly, FA does not always necessitate rigid nutrition interventions. Arguments for and against the FA construct in the context of binge eating have recently been published [15]. Here, the position that FA can be relevant as a clinical entity is presented, and it is suggested that the presence of other psychiatric conditions such as ED, SUD, and posttraumatic stress disorder (PTSD) can be useful in determining if an individual requires targeted nutritional treatment (e.g., abstinence from specific foods). Alternatively, related psychopathology (e.g., chronic dieting based on body dissatisfaction) might indicate an opposite approach (e.g., inclusion of the foods persistently avoided). An understanding of different FA phenotypes may help guide intervention strategies.

### 1.1. Food Addiction Stigma

Alcohol use disorder (AUD) and SUDs were once viewed as individual choices but scientific progress and changing social norms have reduced this stigma. A survey study found that FA is more vulnerable to stigmatization than alcohol and may be perceived as a behavioral rather than substance addiction [19]. This position has favored the term “eating addiction” which has gained some traction and stimulated scholarly debate [20,21]. Meanwhile, FA is becoming increasingly accepted by the lay public, as evidenced by growing numbers of self-perceived food addicts [22,23]. Study participants express a desire to have their perceived condition formally recognized in order to receive more appropriate treatment [24]. While some research suggests that the FA label may increase stigmatizing attitudes [25], other studies show that the FA explanation reduces weight stigma [26,27]. It has also been suggested that while FA reduces externalized stigma, it may increase internalized stigma [28]. Furthermore, believing that certain food products can be addictive has been associated with support for policies intended to curb their use [29]. However, it remains unclear how a better understanding of FA neurobiology can reduce stigma in the context of ED treatment and recovery.

### 1.2. Food Addiction Controversy

Efforts to clear confusion around the FA construct have focused on semantics. For example, some authors have proposed the terms “refined food addiction” or “processed food addiction” to better capture FA as a substance-related disorder [30,31]. This approach targets specific foods which have been identified as addictive, such as chocolate, ice cream, French fries, pizza, cookies, chips, and cake [32]. More importantly, individuals would identify foods to avoid (“trigger foods”) as part of their personal recovery program, which has been endorsed by some 12-Step groups (whereas other 12-Step groups have fixed food plans for all members). These approaches have come under scrutiny because success rates have not been well documented. Meanwhile, some authors posit that saying abstinence is ineffective because people binge when they finally eat sweets is like saying that abstinence from alcohol is ineffective because those with AUD binge after taking the first drink [31]. Another reason “abstinence-based” approaches are criticized is because they often place emphasis on weight loss, which current data suggests is not sustainable over the long term. A study based on a large prospective cohort from the UK suggests that over a 9-year period, the probability of going from obese to normal was 1 in 210 for men, and 1 in 124 for women [33]. Understandably, with such small chances of sustaining weight loss, classic ED treatment has favored targeting dietary restraint rather than weight loss, or the removal of specific foods.

### 1.3. Dietary Restraint

In 2003, Fairburn introduced a transdiagnostic theory of EDs proposing that diagnosis is not relevant to the treatment [34], when binge eating disorder (BED) was in the DSM-IV appendix and not yet an official diagnosis. A core assumption is that dieting precipitates bingeing; therefore, the pursuit of weight loss will be counterproductive with most ED presentations. However, among women with body image concerns (*n* = 1165), weight suppression correlated with future onset of EDs characterized by dietary restriction or compensatory weight control behaviors, but not with BED [35]. In one study of BED outpatients (*n* = 98), 65% reported an onset of dieting prior to their first binge and 35% reported that binge eating preceded their first diet [36]. Thus, dieting may not necessarily be a precursor to all forms of binge eating [37,38]. There is less consensus regarding the pursuit of weight loss in the presence of FA without an ED. Some advocates for reconceptualizing weight as a social justice issue believe that no person at all should engage in weight-loss behaviors. Rather, treatment professionals should target the problem of fat shame in society [39]. Others support the idea that new treatments are needed to address ED pathology and weight loss concurrently [40]. A recent systematic review and meta-analysis found that structured and professionally run obesity treatments are associated with reduced ED prevalence, risk, and symptoms in children [41]. However, because clinicians who work with patients with EDs often discover that the onset of disordered eating follows the first attempt to diet, many prefer a “do no harm” approach. Additionally, many professionals who work with these populations have abandoned weight loss [42] in some cases to avoid being targeted (shamed) by colleagues.

The current standard ED treatment is associated with high rates of relapse and poor long-term outcomes [43,44,45]. It has been suggested that contemporary ED models devote relatively little attention to biological factors driving binge eating, and that changes in the food environment interacting with individual vulnerability are key predisposing risk factors [46]. Newly proposed models suggest that clinicians go beyond a “no dieting” approach for all ED presentations and should incorporate addiction neuroscience [46,47]. Some authors recommend that researchers and clinicians distinguish between flexible and rigid restraint [14]. In some cases, restraint is related to a lower body weight, better weight regulation, and a better diet quality while in others, restraint predicts poor diet, overeating, and obesity [48]. While short-term deprivation increases cravings for avoided foods, long-term restriction results in reduction of food cravings that can facilitate extinction of conditioned responses [16]. Meule states that “the wide-held notion that dieting inevitably leads to food cravings is strongly oversimplified as the relationship between food restriction and food craving is more complex” [16]. This paper explores the nuances of different FA phenotypes, which have not been adequately described.

## 2. Eating Disorders

### 2.1. Bulimia Nervosa and Anorexia Nervosa

The highest prevalence rates of YFAS 2.0 diagnoses have been found in individuals with bulimia nervosa (BN) [49]. The relationship between FA and BMI has been described as non-linear: FA symptomatology can be higher in some underweight groups, in some cases related to compensatory behaviors that maintain lower BMIs [50]. In fact, some studies suggest that when separating anorexia nervosa restrictive type (AN-R) with the binge–purge type (AN-BP), FA prevalence is the highest in AN-BP [51]. Compensatory weight control behaviors in individuals with BN and AN-BP likely dampen the association between FA and BMI [52]. Several key reviews have summarized neurobiological overlaps with BN and SUDs, including dopamine (DA) D2 receptor-related vulnerabilities, structural and functional alterations in the frontal cortex, glutamatergic signaling, and the opioid system [53,54,55]. A recent systematic review of neuroimaging studies on BN and BED found diminished activity in frontostriatal circuits (associated with self-regulation) [56]. Treatment studies suggest that FA most likely improves when BN symptoms remit [57]. In an intervention study among women with BN (*n* = 66), those with higher FA severity at baseline were less likely to obtain abstinence from binge–purge episodes following treatment [58]. Taken together, there is preliminary support for the role of FA in the maintenance of BN via neuroadaptive changes in reward circuits. The challenge is discerning which came first (or would be considered “primary”) in order to conceptualize an effective nutrition strategy. Here, it is suggested that consideration of the temporal sequence of disorder onset can be useful in discerning the truly positive FA “signal” from the falsely positive “noise,” but that no rule can be applied to all cases.

### 2.2. Binge Eating Disorder

It has been suggested that subtyping BED based on psychiatric comorbidity may have important implications for treatment [59]. Researchers question whether the presence of BED vs. FA vs. BED + FA requires tailored treatment approaches [60]. One study suggested that when diagnostic subtypes are considered separately, FA is associated with a poor prognosis in the BED group [61]. It is possible that poor BED outcomes stem from the transdiagnostic assumption that BED patients do not need to emphasize the quality of their food (“it’s not about the food”). Patient interest in weight loss may increase their risk of dropout from nonspecific ED treatment (not tailored to BED) [62]. Different phenotypes in BED are likely related to their dopaminergic response to highly palatable foods. For example, some individuals may develop particular eating expectancies in the face of poor emotion regulation and high anticipatory rewards [63]. The presence of FA may represent a more disturbed group of BED characterized by greater psychopathology [64]. To illustrate, in one study of 788 adults enrolled in BED treatment, shape/weight overvaluation differentiated BED severity more strongly than binge eating frequency [65]. Relatedly, those with heightened body image disturbance are more likely to engage in dietary restraint [66] which may contribute to bingeing as well as FA symptoms. Importantly, fear of being stigmatized predicts worsening FA status over time [67]. Stigma leads to maladaptive eating behaviors, stress, and weight gain [68]. Meanwhile, other findings do not support prevailing models that posit dietary or cognitive restraint as the predominant risk factor in BED [69]. Taken together, there appears a timely need to identify phenotypes of BED that require different treatment strategies, specifically those that would aim to reduce addiction-like eating versus those that would aim to reduce dietary restraint.

## 3. Substance Use Disorders

Given the neurobiological overlap between SUDs, EDs, and FA, considering an individual’s relationship to alcohol, drugs, and other substances such as nicotine and caffeine may be helpful in separating the FA signal from the noise. For example, in a sample of Dutch adolescents (*n* = 2653), symptoms of FA were positively associated with alcohol use, cannabis use, smoking, and sugar intake [8]. Men with heroin use disorder (*n* = 100) had triple the odds of meeting criteria for BED or FA compared to controls [70]. Not surprisingly, FA diagnosis was associated with more severe craving. In a community-based sample of women (*n* = 3756), those with lifetime AUD or nicotine dependence were at higher risk for ED symptoms and diagnoses [71]. Based on the concept of reward dysfunction (reviewed below), it is likely that such ED symptoms represent those that overlap with FA. A recent systematic review and meta-analysis confirmed a higher prevalence of comorbid SUD in binge–purge ED presentations [72]. Among individuals with EDs, the pooled lifetime prevalence of comorbid SUD was 21.9% [72]. In an Italian sample of SUD patients (*n* = 575), the prevalence of FA was 20.2% [73], which is very close to the prevalence of ED estimated in meta-analysis. In a large longitudinal study from Australia, illicit substance users had significant risk of developing recurrent binge eating in addition to, or in place of, their substance use; however, the reverse was not found [74]. This suggests that individuals who engage in dysfunctional food-related behaviors prior to using drugs and alcohol may represent a different phenotype than those who develop addiction-like eating as a result of their drug use. In a study of women in SUD treatment (*n* = 297), a third reported starting drug use (in part) to lose weight and nearly half were concerned that gaining weight could trigger relapse [75]. In a non-treatment sample of drug-using women (college setting), 15.3% reported drug use for weight control purposes [76]. Patients with co-occurring SUD and ED are more sensitive to reward, have more difficulty engaging in goal-directed activity, are more impulsive, and have less access to emotion regulation skills [77]. The assessment of other addictions may prove helpful in determining the underpinnings of a high FA score, specifically as they relate to reward dysfunction and impulsivity. Meanwhile, genome wide association studies have found limited support for shared underpinnings for FA and SUD [78] whereas other lines of research have implicated DA-D2 receptors [79,80,81,82,83].

### 3.1. Reward Dysfunction

Convergence from neuroscience findings with case reports from the field have made clear that dysregulation of DA function is important for reward-related processes driving substance-seeking behavior [84]. Many authors have speculated that the bidirectional association between food and alcohol/drug dysfunction represents an “addiction transfer” [85] which has been supported by many studies of bariatric patients [86]. In a large twin study from the Netherlands, genetic factors explained 48% of the variation in high sugar consumption (52% explained by unique environmental factors), suggesting that neuronal circuits underlying the development of addiction and obesity are related, possibly due to DA receptor dysfunction that lead to difficulties resisting rewarding stimuli [87]. DA contributes to addiction and obesity through its differentiated role in reinforcement, motivation, and self-regulation [88]. In addition to deficiencies identified at striatal DA-D2 receptors [89,90], individuals with obesity and BED have widespread reduction in binding at mu-opioid receptors (MOR) [91]. The mesolimbic dopaminergic circuit is clearly affected by both highly palatable foods and diet-induced obesity similar to exposure to drugs of abuse [92]. Recent review articles have discussed highly processed foods (often high in glycemic index) as impacting neurohormonal and inflammatory signaling pathways in ways that create a vicious cycle of impulsivity, compulsivity, FA, and EDs [93,94,95]. The Regulatory Model of Addictive Vulnerability (RMAV) proposes that susceptibility to addictive disorders is linked to how well an individual’s regulatory system responds to challenges, also referred to as allostasis [96]. According to these authors, both obesity and drug addiction are examples of major disorders characterized by dysregulated control systems. To date, this information has not been integrated into mainstream ED treatment programs, possibly because the translation of these findings contradicts the popular assumption that “it’s not about the food.” No trials have been conducted using strategies designed to reduce reward-based eating in patients with EDs; however, there are data suggesting that medications commonly used in SUD treatment (naltrexone/bupropion) in conjunction with lifestyle changes can reduce FA severity among those with BED [97]. Prospective research is needed to determine if reduction of highly palatable foods can improve reward dysfunction in people with FA.

### 3.2. Impulsivity

Impulsivity can be separated into attentional (inability to focus attention or concentrate) and motor (acting without thinking). In a sample of individuals with obesity presenting for bariatric surgery (*n* = 193), FA emerged when both attentional and motor impulsivity levels were elevated [98]. Impulsivity has been identified as a key shared mechanism between BED and addictive disorders [99]. For example, patients with EDs who have problems pursuing tasks to the end and focusing on long-term goals are more likely to develop addiction-like eating patterns [100]. A recent systematic review reported that across 45 studies, impulsivity was consistently associated with FA [101]. Given that FA has been reported as a mediator between impulsivity and obesity [102], it is possible that in certain susceptible individuals, ED behaviors develop along this trajectory to suppress unwanted weight. In a recent study of 145 patients with EDs, those with alcohol and drug abuse symptoms represented a specific phenotype characterized by greater impulsive personality, emotion dysregulation, and problems with executive functioning [103]. Among male military veterans (*n* = 106), impulsivity moderated the relationship between PTSD symptoms and alcohol consumption [104], with relevance discussed in more detail below. In addition to higher levels of impulsivity, individuals with FA are more likely to report a family history of addiction [105] which may be consistent with genetic underpinnings. While addiction research has primarily focused on the mesolimbic dopaminergic projection, impulsivity has also been linked to the serotonin system. Several genetic studies have linked polymorphisms at 5HTTLPR (codes for serotonin transporters) to higher levels of impulsivity among individuals with BN [106,107], with associated aberrations of serotonergic functioning being exacerbated by early life adversity (ELA) [108,109,110]. However, meta-analysis has linked the 5HTTLPR allele strongest to AN [111] and to date no study has linked this polymorphism to FA. Taken together, assessment of impulsivity in conjunction with assessment of SUD may prove beneficial in separating the FA signal (true positives) from the noise (false positives), but more research is needed before the impulsivity construct can be used in predicting the clinical utility of FA.

## 4. Trauma and PTSD

The Adverse Childhood Experiences (ACE) study from 1998 highlighted 5- to 10-fold increase in the risk of AUD and SUD following exposure to four or more ACEs in the first 18 years of life [112]. As traumatic stress affects a variety of brain structures and functions, ACEs impact a variety of functions and behaviors therefore have been determined nonspecific [113]. ACEs captured by the various forms of the questionnaire are referred to as a form of ELA; however, it is important to acknowledge there are many other measures used, such as the childhood trauma questionnaire (CTQ) [114,115]. Why is it that some individuals exposed to childhood trauma have a heightened risk for psychiatric disorder while others demonstrate resilience over the lifespan? The Theory of Latent Vulnerability suggests that one’s genotype interacts with an adverse environment to create a neurocognitive phenotype, characterized by changes in reward processing (DA), threat processing (amygdala), and memory processing (hippocampus) [116]. One pathway which might in part explain the enduring biological impact of adversity is inflammation. A longitudinal study of adolescent girls (*n* = 147) showed that ELA was associated with greater odds of displaying a proinflammatory phenotype, generating low-level non-resolving inflammation (higher levels of IL-6 and decreased sensitivity to cortisol) [117]. Meta-analysis has linked childhood trauma to cognitive deficits, with the greatest deficits among those with a PTSD diagnosis [118]. If any abuse is identified in children or adolescents, it is likely they have also previously experienced, are currently experiencing, or are at risk for experiencing additional forms of abuse [119], which highlights the importance of social and environmental factors in a biopsychosocial model. While it is outside the scope of this review to clearly distinguish between trauma, adversity, chronic stress, and PTSD, the importance of these events early in life are emphasized as increasing risk for various addictions. Important for the understanding of trauma is that events be differentiated from their effects, which varies based on the experience of the individual [120]. Exposures by themselves do not define PTSD.

### 4.1. Addictions

Functional magnetic resonance imaging (fMRI) studies have linked ELA to blunted subjective responses to reward-predicting cues and dysfunction in the left basal ganglia regions implicated in reward-related learning and motivation [121]. Other neuroimaging studies have indicated an increase in dopamine transporter (DAT) density in PTSD, which may reflect a higher DA turnover among trauma survivors [122]. Both an increase in the number of traumatic events early in life and an increase in levels of perceived stress were associated with a higher ventral striatal DA response to amphetamine [123]. This evidence supports the biological embedding hypothesis [124] which links ELA to addictive behaviors [125]. ELA can be viewed as nonspecific because it predisposes individuals to a wide range of addictive behaviors. For example, in a sample of healthy young adults (*n* = 200), greater lifetime stress exposure was related to increased impulsivity and FA [126], suggesting that food is a predictable go-to for self-medication [127]. A nationally representative sample of young adults’ (*n* = 10,813) exposure to multiple forms of maltreatment predicted excessive sugar sweetened beverage consumption [128]. Not surprisingly, childhood physical abuse and childhood sexual abuse both increase risk for FA by approximately 90% [10]. Higher numbers of PTSD symptoms predict increased prevalence of FA and the strength of this association increased when symptom onset occurred at an earlier age [11]. In this study (*n* = 49,408), the PTSD-FA association did not differ substantially by trauma type, suggesting that all forms of ELA impact reward-related behaviors. Among overweight/obese women (*n* = 301), the association between FA and childhood trauma was significant after controlling for potential confounders such as socioeconomic status [129]. Compared to women with no addictions, women with FA and with SUD endorsed more depression and PTSD symptoms and had more difficulties with goal-directed behaviors, acceptance of emotions, and impulse control [130]. Taken together, exposure to trauma significantly increases risk for FA and therefore should be considered during comprehensive psychiatric/psychological and nutrition assessments.

### 4.2. Eating Disorders

All forms of childhood maltreatment are associated with all forms of EDs, although some more than others [131]. The odds of BED following maltreatment is consistently higher than AN [132], yet meta-analysis confirms that, when pooled, the risk for BN is the highest [133]. Given the link between early life trauma and addictive disorders, it is likely that FA is an important pathway on the trajectory toward binge-type EDs. Meanwhile, there are also associations from ELA to EDs which may not include FA as a mediator, capturing a different phenotype with different treatment implications (discussed further below). In one sample of adult patients with EDs from Sweden (*n* = 853), a quarter had a lifetime diagnosis of PTSD [134]. Other estimates suggest traumatic events occur in over a third of adolescents in ED outpatients (*n* = 182), and the prevalence is highest among those with BN [135]. A recent review article suggests that the prevalence of comorbid PTSD and EDs ranges from 9% to 24% [136], although estimates are consistently higher when looking specifically at binge-type EDs [137,138,139,140,141]. Several studies have suggested that adverse events experienced early in life predict binge eating symptoms in both men and women [142,143,144]. Evidence consistently shows that ED symptoms such as anxiety and depression are more severe among those with a PTSD diagnosis [145], and those with the dissociative subtype are even worse off [146]. Some authors believe that it may be helpful to modify current ED treatments to better address the overlapping risk among EDs and obesity among those who have been exposed to trauma [147]. Others posit that any effort to lose weight or restrict food will only make eating problems worse [66,148,149]. In this paper, our aim is to resolve this debate by presenting three clinical vignettes that are fictional yet rooted in extensive clinical experience.

## 5. Other Psychiatric Diagnoses

### 5.1. Depression

Multiple lines of evidence suggest that individuals with FA have more depressive symptoms than controls [60,105,130,150,151,152]. A recent review of studies using YFAS identified depressive symptoms as a clinically relevant correlate [49]. Meta-analysis suggests that FA is significantly correlated to depression (r = 0.459) [4]. The directionality of this relationship remains less clear. While it is likely that depressed individuals may turn to highly palatable foods to alleviate negative affect, several recent studies have suggested that a low-quality diet increases depressive symptoms. A cross-sectional study from the US (*n* = 16,807) suggests that intakes of total fiber, specifically from fruits and vegetables, was inversely associated with depressive symptoms [153]. In a French cohort of adults (*n* = 3523) with mean follow-up 12.6 years, diets high in anti-inflammatory properties prevented depressive symptoms, particularly among men, smokers, or physically inactive individuals [154]. In a Spanish cohort of graduate students initially free of depression (*n* = 14,907), followed for a median 10.3 years, participants with the highest consumption of ultra-processed foods had the highest risk of developing depression, particularly among those with low levels of physical activity [155]. Another large French cohort of adults followed for a mean of 5.4 years demonstrated a positive association between ultra-processed food and the risk of incident depression [156].

While there appears to be preliminary support that low-quality diets can lead to depression, there is also evidence that high-quality diets can mitigate or reverse depressive symptoms. A randomized controlled trial (RCT) showed that dietary support (nutrition counseling by a dietitian) for 12 weeks can improve symptoms [157]. Systematic review and meta-analysis suggest that the most compelling evidence for reducing incident depression exists for the Mediterranean diet, known for its anti-inflammatory properties [158]. Since depressive disorders correlate and cross-associate with EDs, SUDs, and PTSD [138,159,160] the diagnosis may prove important for informing nutrition treatment. While the presence of depressive symptoms might not be informative of whether or not an individual has an actual addiction to food (the signal), it is worth considering the potential role of dietary intake in contributing to symptoms. For example, for a person engaging excess consumption of highly palatable foods, it may be worth experimenting with dietary manipulation before psychiatric medication. Further, if a patient receiving ED treatment with the “all foods fit” philosophy is consistently eating low-fiber foods (“intuitively”) and has non-resolving depressive symptoms, it may indicate that a more targeted nutrition strategy is warranted.

### 5.2. Anxiety

In addition to reducing depressive symptoms, the Mediterranean diet may also be helpful in reducing the odds of anxiety [161]. However, a recent meta-analysis of RCTs concluded that no effect of dietary interventions is observed for anxiety [162]. Meanwhile, anxiety disorders show a dose–response association with worsening diet quality [163]; however, directionality remains unclear. Omega-3 fatty acids have been investigated for their role in anxiety disorders, but results are inconsistent, and data are too sparse to draw conclusions [164,165]. The role of gastrointestinal microbiota has also received attention as a potential mediator linking diet quality to anxiety symptoms [166,167,168,169]. Meanwhile, anxiety has been significantly correlated with FA (r = 0.483) [4] which is not surprising given the strong associations between anxiety and SUDs [170]. Anxiety is a well-established risk factor for EDs, SUDs, and PTSD. A recent systematic review suggested that anxiety may mediate the association between PTSD and SUD [171]; therefore, it is biologically plausible that anxiety is on the pathway from ELA to FA, though this has not yet been shown. Among treatment-seeking youth (*n* = 490), social anxiety predicts binge eating [172]. A rodent model suggests that consumption of a Western diet may lead to long-lasting damage to fear neurocircuitry, particularly during adolescence [173]. In an Australian sample of adults (*n* = 1344), anxiety sensitivity predicted severe FA [174]. The association between FA and current anxiety disorders has also been reported in bariatric surgery candidates (*n* = 128) [175]. One study suggested that irrational beliefs may be the source of the anxiety associated with FA [176]. Among adult females with anxiety (*n* = 51), impulsivity predicted higher intakes of sugar and saturated fat [177], which is consistent with reports of “comfort food” consumption when under stress [178]. Taken together, the presence of an anxiety diagnosis by itself is not likely to be informative of an FA phenotype; however, it may prove beneficial to discern between anxiety that is symptomatic of PTSD/ELA and generalized [145,179] versus other forms of anxiety such as body image disturbance or dysmorphia (likely indicative of dietary restraint). Either way, FA treatment should include positive anxiety management and coping skills [46].

### 5.3. Attention Deficit Hyperactivity Disorder (ADHD)

ADHD is defined by inattention and/or hyperactivity and is often characterized by impulsivity. Both ADHD and SUDs are characterized by choice impulsivity [180]. There are recent data to support the possibility that choice impulsivity in ADHD results from substance misuse [181]. A popular explanation for the association between ADHD and EDs is that impulsive behavior generates the disordered eating [182,183]. Both conditions rely on a dopaminergic signaling which makes their cooccurrence a logical comorbidity [182]. However, the evidence reviewed above suggests that among those with EDs, impulsivity is also linked to serotonergic genes (i.e., 5HTTLPR). In both sexes, binge eaters have significantly higher prevalence of ADHD [184]. In a nationally representative sample of adults in the US (*n* = 4719), only the association between ADHD and BN remained significant after confounders were adjusted for [185]. In a sample of patients with obesity (*n* = 105), adult and childhood ADHD were significantly associated with self-reported FA and binge eating severity [186]. In a recent study of 136 patients with EDs, a positive screen for ADHD related to worse eating symptoms and the presence of high ED levels contributed to treatment dropout [187]. Meanwhile, a large genome-wide association study suggests that higher BMI increases risk of developing ADHD but not the other way around [188]; therefore, directionality remains unclear. It is worth noting that stimulant medications (i.e., amphetamines) often used in the treatment of ADHD can have an impact on appetite, with suppressing effects while using and rebound appetite when not. In one study of undergraduate students (*n* = 705), nearly 12% reported using prescription stimulants to lose weight [189] however these students were not diagnosed with ADHD. Lisdexamfetamine, which is FDA approved for ADHD as well as BED, works by enhancing dorsofrontal cortex function [190,191]. Meanwhile, it remains unclear whether ADHD symptomatology (e.g., impulsivity) or the effect of ADHD medications (including their discontinuation) drive the potential association with FA. Research is needed to test long-term outcomes of lisdexamfetamine on BED + FA. It may prove important to consider ADHD when conceptualizing FA phenotypes in the context of nutrition strategies.

## 6. Clinical Vignettes

### 6.1. Phenotype A

Alma is a 23-year-old, single Columbian female who grew up in a troubled household. Her parents were not married but lived together off and on until she was 10, when her father left. Alma had two older brothers from a different father. Her mother divorced her first husband when the two boys were two and four, right before she met Alma’s father, who attempted to parent all three children. However, Alma’s father struggled with a severe alcohol use disorder and would be absent for days at a time. Alma’s brothers never accepted him as a parent figure. When Alma’s father was gone completely, she began to seek attention from boys by playing sports with her brothers. She became athletic and thrived in volleyball. As a freshman in high school, she joined the volleyball team and made many friends among the athletic crowd. Alma loved to play sports and cook food, often referring to herself as a “foodie”, known for baking Columbian desserts for her teammates. She grew close to the assistant coach of the girls’ volleyball team who had a reputation for drinking with the students after games. Alma never drank alcohol or did drugs because of what she had seen them do to her father. She swore she would never smoke a cigarette. One night the male assistant coach offered to take Alma home and kissed her in the car. Alma felt confused but kept the secret to herself. During the summer after her freshman year, she agreed to visit his home where she was coerced into sex (statutory rape). Alma kept it a secret due to shame, but people close to her knew something had changed.

Alma did not return to the volleyball team for her sophomore year. She became promiscuous with several seniors at her school and some of her brother’s friends. She lost interest in sports completely and began binge eating on highly processed foods at night to help her sleep. She often skipped breakfast but never engaged in any compensatory behaviors. Her academic performance began to decline, and she started drinking coffee, soda, and energy drinks throughout the day. Between her sophomore and senior year, Alma gained forty pounds, meanwhile experimenting with a few popular diets that never stuck for more than two days. When she graduated from high school, she moved in with her boyfriend who was 30 years old, owning a small shop that sold electronic cigarettes. Alma began vaping daily. However, she stuck by her commitment to never smoke a cigarette, drink alcohol, or use drugs. They mostly ate fast food and ordered take-out together, but Alma would frequently bake and cook. Alma got a job as an office administrator and slowly stopped responding to texts from her mother and brothers. She never posted any pictures on social media because she did not want people to see how much weight she had gained. Eventually, she learned that her boyfriend was cheating on her and moved back in with her mother after he told her that she had to leave. At this point the mother brought Alma to a psychiatrist for an evaluation, but Alma did not tell the doctor about the rape, in part because she was amnestic for the memories. The doctor identified complex PTSD and a dissociative disorder, prescribing sertraline and trazadone. She was referred to an outpatient psychotherapist who began working with her twice per week, at which point Alma opened up about the sexual abuse. When the trauma therapy started, the bingeing escalated significantly. At this point, Alma was referred to a registered dietitian nutritionist (RDN) specializing in mental and behavioral health. Several assessment tools were administered, including the eating disorder examination questionnaire (EDE-Q) which indicated an absence of dietary restraint and the YFAS 2.0 where she met criteria for severe food addiction (BMI = 36.8).

### 6.2. Phenotype B

Jeffrey is a 27-year-old, single, half-Taiwanese, half-white male who grew up in a wealthy household as an only child. His father was a cardiologist with mild undiagnosed obsessive compulsive personality disorder (OCPD) and, before getting married, his mother was a swimsuit model with a long history of dieting. Both parents ran marathons together and revered the thin ideal, frequently making negative comments about fat people. Jeffrey’s parents were very adamant about their son playing sports and at one point hired a running coach for him. Jeffrey spent a lot of time with nannies and babysitters, including extended periods when his parents vacationed without him. When Jeffrey was nine, he was sent to a psychiatrist for behavioral problems where he was diagnosed with ADHD and prescribed dextroamphetamine/amphetamine. Jeffrey was a straight-A student with the help of the medication and several tutors. Jeffrey also began running with his parents by age 13. His mother told Jeffrey to only eat carbohydrates before, during, and after a run, and to avoid them at all other times. Jeffrey completed his first marathon at age 15 and wore the medal at his private high school the next day, where he was photographed for the cover of the school magazine and website. One of his mother’s friends provided an opportunity for Jeffrey to model for a large international fashion company and, at age 17, he was on a billboard in his 6′2” frame with his shirt off. His agent helped him build a following on Instagram, and Jeffrey began to spend more time exercising to prepare for photo shoots. The agreement was that Jeffrey could continue to model as long as he went to college and earned decent grades. He went to a private university as a communications major.

As a freshman in college, Jeffrey was switched from mixed amphetamine salts to a high-dose lisdexamfetamine and was also prescribed clonazepam for anxiety. Jeffrey never told anyone that he spent up to an hour each day looking in the mirror, obsessed with the fat on his abdominal area. During his junior year Jeffrey did a “Freeze the Fat” procedure with the physician who had done several cosmetic surgeries on his mother. He was very disappointed with the results. As Jeffrey continued to get modeling jobs, his performance in school began to decline. Jeffrey continued to run 10 miles per day on a treadmill in the gym at his apartment complex. He also did 30 minutes of abdominal workouts daily and rarely missed a workout. Jeffrey cut out all grains and dairy from his diet and only got small amounts of carbohydrates from fruits, starchy vegetables, and occasionally beans. Jeffrey frequently made negative comments about processed food and did not like to eat at restaurants. Jeffrey had several short-lived relationships with women, but as he became increasingly concerned about his appearance, he lost interest in dating. Jeffrey sought out an RDN to help him lose his stubborn abdominal fat; the dietitian determined that his BMI was 17.2 and contacted the psychiatrist to discuss potential body dysmorphic disorder. The psychiatrist told Jeffrey that being on a stimulant was contraindicated at such a low BMI, discontinuing the lisdexamfetamine and starting him on fluoxetine. At this point Jeffrey starting bingeing on carbohydrates and would “run it off” in his apartment gym, even if it was late at night. One evening, Jeffrey rolled his ankle and due to the sprain was told not to run for several weeks. Jeffrey purged for the first time after ordering Chinese food and eating a whole container of fried rice. He hated the experience of purging but continued to engage that behavior whenever he ate carbohydrates. He began to order food delivery and would often flush the food down the toilet because once he retrieved it from the trash. At the request of the psychiatrist, Jeffrey made another appointment with the RDN and obliged because he was feeling very depressed. Several assessment tools were administered, including the EDE-Q, which indicated high levels of dietary restraint, and the YFAS 2.0, where he met criteria for moderate food addiction (BMI = 17.0).

### 6.3. Phenotype C

Whitney is a 30-year-old, single, white female, and the oldest of three daughters to happily married parents who had no formal psychiatric diagnoses. Her mother was a professor at a small university and her father a certified public accountant who became obese in his late 40s. Her two sisters looked up to Whitney who was quite popular in middle school. Whitney got a lot of attention from boys but found herself attracted to women. In high school, she had a girlfriend for two years and the relationship made a lot of people in school uncomfortable. Despite becoming somewhat of an outcast, Whitney got good grades and was accepted into a local college as a sociology major. She was passionate about gender studies and started her own blog about sexuality. For a class assignment, Whitney filled out the ACE measure and scored a zero. Some of her friends began frequenting rave parties and taking ecstasy; Whitney fell in love with this culture. Whitney attended the Burning Man festival each year during college and graduated with honors. Whitney’s family was proud of her and she maintained successful relationships with her parents and siblings. Everyone in the family loved Whitney’s girlfriend. After college, they opened a business selling handmade jewelry, which was a big success in the Burning Man crowd. They eventually moved into a condo together and had two dogs who they took with them on long hikes at nearby trails.

At age 26, Whitney was hit by a drunk driver which killed her partner, witnessing her take her last breath at the scene. Whitney shattered her femur along with several other minor fractures and was in the hospital for two weeks, completely devastated by the loss of her lover. Despite having family by her side constantly and a doctor who gave a positive prognosis of her recovery, Whitney began to express suicidal ideation. She was on heavy doses of opioid medications and everyone assumed her mental health would improve after discharging from the hospital. However, upon returning back to her parents’ house, it was obvious that Whitney had PTSD. She was prescribed oxycodone for pain management which she quickly became dependent on. The family found a therapist to do Eye Movement Desensitization and Reprocessing (EMDR), but Whitney would often show up sedated from the medication and the work was not productive. After several months of both physical and emotional therapy, the doctor took her off opioids and prescribed gabapentin. However, Whitney was able to purchase oxycodone on the street and within a matter of months she was buying heroin (which she smoked rather than injected) because it was much more affordable.

Whitney went into her first treatment center for heroin use disorder at age 29. She was prescribed gabapentin, buspirone, methocarbamol, venlafaxine, and quetiapine. During this time, she developed a strong preference for sweets and would eat several bags of candy daily. It was normal for people in her rehab to smoke cigarettes and drink sugary energy drinks, so she did the same. For the first time in her life, Whitney began gaining weight and by six months sober had gained 30 pounds. At this time, she was in a sober living house, which provided restaurant-style catered food. One of her roommates with severe bulimia nervosa taught Whitney how to induce vomiting to lose unwanted pounds. Whitney had never struggled with body image issues until she was sober and on several medications. She quickly learned that purging was quite soothing and began vomiting daily, yet never tried any specific diets. Sometimes she would purge “healthy foods” simply because it felt relieving to do so. When these behaviors were discovered, her sober living required her to attend an ED outpatient clinic where she was instructed to eat three meals, two snacks, plus dessert every single day. After reaching her highest weight, Whitney relapsed on heroin and, within two weeks, was back in detox, where she began bingeing on ice cream, candy, and grilled cheese sandwiches. Whitney was referred to an RDN specializing in mental and behavioral health. Several assessment tools were administered including the EDE-Q which indicated moderate dietary restraint and the YFAS 2.0 where she met criteria for severe food addiction (BMI = 26.4).

## 7. Discussion

Phenotype A is a clear example of an FA signal (true positive) not blocked by the noise of dietary restraint (false positive). With an absence of restrictive eating there is little convincing evidence that the addiction is driven by dieting, a relic of internalized weight bias, or other forms of compensation often generated by socially constructed forces such as weight stigma. This case illustrates the link between ELA and FA which is likely mediated by biological mechanisms including altered DA signaling [121,122,192,193,194]. FA has been independently associated with exposure to early life sexual abuse [195]. Alma’s father has AUD, which can be useful in evaluating biological susceptibility to addictive disorders. Furthermore, Alma was a “foodie” prior to the rape incident, suggesting that her tendency to seek highly palatable food may have a genetic basis as well as linked to her early psychosocial environment (and cultural background). The traumatic event did not create the FA but rather exacerbated her symptoms and severity by creating dependence on food for self-medication. Her weight gain may have reinforced incentive salience assigned to food stimuli [196]. Alma did not develop alcohol or drug problems due to important social factors during her upbringing where she witnessed the devastating impact they had on her father. While many people with high susceptibility to addictions struggle with multiple substances, Alma did not cross-addict into intoxicating substances; however, her clinical history indicates evidence of both caffeine and nicotine use disorders. Based on our clinical experience, we have observed that some (not all) of the most severe FA cases develop addictions to food with limited cross-addictions. We recommend considering cross-addiction in discerning the signal from the noise (true versus false positives); however, the absence of cross-addiction does not always indicate a weaker signal. In fact, in some cases it may be more severe because other addictive substances fail to compete with the experience of food. As this phenotype exemplifies FA, it suggests that FA-informed nutritional strategies may be warranted as well as safe (low risk of developing an ED). We suggest that strategies be assessed individually rather than a “one size fits all” approach.

Phenotype B is an example of how “noise” can muddle the FA signal and produce a false positive. In this case, dietary restraint is clearly driving the FA symptoms. Jeffrey is a classic example of how thin ideals can be perpetuated by the family system, become internalized, and eventually become pathological. Jeffrey’s modeling career appears influenced by his mother’s history of modeling and related social network. There is no AUD/SUD in the immediate family, which can be helpful information when assessing potential FA (reward dysfunction). The fact that Jeffrey never struggled with AUD/SUD is also informative. However, the ADHD diagnosis indicates the potential for higher levels of impulsivity and the stimulant medications may have appetite dysregulating effects [182,189,197]. When Jeffrey was taken off lisdexamfetamine after clinical concerns about under eating and body dysmorphic disorder, the loss-of-control eating began. Binge eating was accompanied by compensatory exercise behaviors, indicating classic ED pathology. The period of time when Jeffrey was depressed and experiencing heightened conflict around food is likely the source of the increased FA scores. He was previously successful in restraining himself with food but eventually lost his ability to do so, which is not uncommon over longer periods of time [34]. This case study exemplifies how an underweight individual can have elevated YFAS scores; however, the diagnosis of moderate FA is not informative for treatment. Rather, FA is a product of restrained eating stemming from body dissatisfaction. Therefore, intervention strategies should focus on dietary inclusion rather than exclusion, targeting the underlying psychological and family system issues rather than focus on avoiding specific foods (e.g., carbohydrates).

Phenotype C is an example of a case that could be interpreted as an FA signal (true positive), or as a more classic ED presentation (false positive), depending on the training (and bias) of the practitioner. Whitney had no evidence of ELA, although did experience adversity in high school. However, she was resilient to this exposure. She did not develop SUD early on despite regularly using “party drugs” (methylenedioxyamphetamine or MDMA). However, following her serious injury and PTSD warranting opioid medications, she developed an opioid use disorder, implicating trauma as an important risk factor. Once becoming sober and being prescribed several medications, her PTSD symptoms and reward-seeking behavior led to FA and associated distress. Evidence of cross-addiction can be found with caffeine and nicotine. The ED developed after the FA and weight gain, eventually contributing to her relapse with heroin. While body dissatisfaction initially drove the ED behaviors, there was no history of dietary restraint before the SUD. In this case, FA preceded the dietary restraint, meaning that the signal existed before the noise. It is likely that practitioners who do not endorse the clinical utility of FA will observe the PTSD–ED connection and ignore the contribution of addiction-like eating into Whitney’s assessment and treatment plan. The best direction for nutrition treatment is not clear and could be effective in multiple ways as long as Whitney had “buy-in” with adequate clinical and social support. Regular inclusion of highly palatable foods appears to be the safest course in order to prevent further progression of restriction and/or purging. However, it is possible that this approach can increase risk for SUD relapse if Whitney continues to gain weight and is unable to accept her body at a higher BMI [75]. Meanwhile, some might argue that FA-informed nutrition strategies that reduce reward-based eating may increase risk for SUD relapse by depriving the brain of DA that it has been conditioned to get from comforting foods. This has been widely endorsed by a “first things first” message from *Alcoholics Anonymous*, suggesting that sweets and chocolate are helpful in early recovery [198], but data to support this are lacking. This is another example of how a nutrition strategy is best assessed on an individual basis. Importantly, the treatment team should be on the same page since consistent messaging from providers appears critical [47].

The three clinical vignettes bring attention to heterogeneity that is possible given an FA diagnosis using the YFAS 2.0. All three cases meet criteria for FA (A and C are severe while B is moderate). However, a comprehensive biopsychosocial assessment identified divergent phenotypes that may warrant different nutrition interventions. While no trials have been reported using targeted nutrition interventions for FA, several studies have shown that non-diet approaches such as intuitive eating can be effective in reducing dietary restraint [199,200,201]. The present review suggests that it would be effective to identify FA phenotypes based on the presence of other psychiatric disorders such as ED, AUD/SUD, PTSD, depression, anxiety, and ADHD as part of a comprehensive biopsychosocial assessment, and to assign nutrition treatment based on the relative strength of the FA signal amidst the noise (true versus false positive). Recent studies have identified different phenotypic characterizations of the FA construct [60,61,202,203]. However, Table 1 suggests a guide for clinicians to consider in settings where the use of extensive validated instruments is not always practical.

## 8. Interventions for Food Addiction

There are few successful interventions for reducing FA in the literature. Likewise, there are no articles describing effective interventions for the treatment of obesity in individuals with a history of ACEs [220] which is likely mediated by FA [9,192]. In a study of 60 women, 12-Step self-help groups for compulsive eating have been shown to reduce anxiety and depression, but not FA [221]. A 14-week group lifestyle modification program including caloric reduction (*n* = 178) significantly reduced addictive eating behaviors [222]. A 6-week integrative group for weight management (*n* = 51) reduced FA from pre to post, with strategies such as mindful eating, keeping a food diary, carrying out an exercise plan, regular weigh-ins, and planning for social eating [223]. In a study of 47 different internet sources, self-perceived sugar addicts shared actional strategies that worked for them, including avoidance, consumption planning, environmental restructuring, professional and social support, addressing underlying issues, and urge management, among others [23]. Qualitative interviews have found that the YFAS does not adequately assess social and situational cues for overeating [224]. Importantly, interventions aimed to reduce weight and FA scores generally do not have strategies in place to mitigate progression into disordered eating. This highlights the difference between dietary restraint that can be helpful for some versus pathological for others. Some authors recommend that if an ED is present in addition to FA, clinicians should first provide evidence-based treatments for those conditions [225]. Notwithstanding, it is worth repeating that among women with BN, patients with higher FA severity at baseline were less likely to obtain abstinence from bingeing/purging episodes after treatment [58]. Thus, it is being suggested to view EDs with co-occurring FA on a continuum rather than as discrete conditions, using the eight-step process as a guide, rather than simply to dichotomize inclusive vs. exclusive nutrition strategies.

It is well established that earlier intervention is beneficial for addressing ED pathology [226]. With respect to reducing FA symptoms and severity, it appears that earlier intervention matters, given that ELA does not lead to obesity immediately but develops over time [227,228]. Recently, there have been recommendations for interventions among adolescents that promote executive functioning in the context of salience and reward processing [196]. Among adolescents with obesity (*n* = 18), an FA-informed mobile health (app) intervention reduced zBMI in a more cost-effective manner than the in-clinic intervention, and there is currently an RCT underway in a larger sample using this approach [229,230]. It has been suggested that the more interactive, engaging and person centered a mobile health treatment is, the more appealing it will be to those suffering from compulsive overeating [231]. Meanwhile, many people with more classic ED training may view these apps as a causal factor to ED pathology, either in the short term or over the life course. It seems that until researchers and clinicians determine who is a good candidate for an FA-based nutrition intervention, FA science will continue to stimulate disagreement. In the meantime, recommended treatments might include abstinence from trigger foods, deliberate inclusion of health-promoting foods, interventions that target impulsivity and habitual patterns of responding, anxiety management, coping mechanisms, positive social connections, spirituality, and deterrence of maladaptive compensatory behaviors [46].

## 9. Summary

The DEFANG (2017) was the first effort to disentangle FA from more classic ED pathology by incorporating the presence or history of SUD into the nutrition intake process [47]. The current review using three clinical vignettes extends that work by adding trauma and PTSD history, particularly early in life, as well as histories of depression, anxiety, and ADHD to guide treatment. The aim in including psychiatric diagnoses, self-report, scores on validated measures, or even informal assessment (clinical intuition) is to reduce the potential for false FA positives (enhance specificity and sensitivity). We have recognized dietary restraint as a primary contributor of “noise” in the FA signal. Failure to consider restrictive eating patterns is an important criticism of FA that has led many ED professionals to reject the construct altogether [15]. The eight-step process outlined in Table 1 might improve the FA assessment process and help clinicians further integrate FA into ED treatment protocols. Currently, most EDs are treated with an inclusive nutrition strategy aimed to reduce fears around food and desensitize individuals to highly palatable foods through regular consumption. Meanwhile, standard ED treatment is associated with suboptimal results, possibly because existing treatments sometimes fail to recognize impulsivity as part of the eating pathology [232,233]. Our clinical experience suggests that failure to recognize/treat trauma/PTSD is a major contributor to poor outcomes. We have suggested that the proper interpretation of an FA diagnosis may improve treatment for those who would benefit from a different nutritional approach, such as excluding problematic foods like added refined sugars [234]. Evidence supports the validity of FA as a diagnostic construct, particularly as it relates to foods high in added sweeteners and refined ingredients [235]. Similar to how SUD patients exhibit different patterns of abuse, patients with FA may have very different behavioral characteristics such as those that binge versus those that do not. Tailor-made hybrid models between inclusive and exclusive approaches have been useful in our clinical experience but have yet to be formally described or tested. These nutrition interventions usually require some trial-and-error and are best done under the supervision of an RDN and a psychiatrist/psychotherapist who understands EDs, FA, SUDs, trauma, and the associations with other psychiatric diagnoses described herein. A multidisciplinary team approach can be helpful, but it is essential that all team members understand the science of FA.

Emerging data on FA may contribute to reduced stigma around body weight, by clearing up confusion and controversy around why humans consume food beyond physiological need. Better terminology will be important to progressing FA science, with several authors proposing new descriptors such as “food use disorder” [236] among others described earlier. The area of greatest controversy surrounding FA appears to be in those with clinically significant EDs, particularly those with purging behaviors where FA symptoms can become elevated [51]. An understanding of the SUD recovery culture including harm reduction may be useful in helping clinicians integrate FA into ED treatment. However, it will be very challenging to implement divergent nutritional strategies in residential treatment settings where there is comparison on the unit and heightened interest in each other’s food plans. Currently, exclusive nutritional strategies might be best conducted in an outpatient setting. These strategies may run the risk of exacerbating restrict–binge patterns, therefore should be supervised by clinicians experienced in EDs as well as in detecting reward dysfunction (through examination of cross-addictions) and impulsivity (with respect to food and other behaviors). Treatment professionals should be aware of the various pathways in which ELA and PTSD can become biologically embedded and alter human physiology. ELA increases vulnerability for FA and obesity later in life [192] and highly palatable foods can become a way to distract from disturbing and intrusive trauma-related thoughts [237]. When food alters DA circuitry, efforts to moderate become more difficult and “intuitive eating” can feel impossible. Trauma-informed care should be applied to recovery systems and providers servicing EDs [238,239].

### Future Directions

A large prospective study of individuals meeting criteria for FA separated into a restricted diet group (excluding identified trigger foods) and a non-diet group (including challenging foods) would be informative, timely, and warranted. However, given the heterogeneity associated with FA described herein, it could be more effective to categorize FA phenotypes before implementing nutrition-related treatment. Future research using exclusive nutrition strategies might exclude participants revealing moderate or high levels of dietary restraint in order to assess risk for progression from exclusion into disordered eating, as well as into “orthorexia” [240]. The inclusion of orthorexia into future versions of the DSM might prove useful when conceptualizing treatment strategies and research related to FA. It would be valuable to analyze the benefits vs. risks in reducing reward-based eating in those who meet criteria for FA. Research on medications commonly used on patients with SUD for patients with FA is also needed. There is a growing interest in genetic risk for EDs and it would be valuable to know if genetic counseling would be of benefit [241]. More data on food insecurity and other forms of deprivation as predictors of FA may also elucidate the link between undereating and overeating. The temporal sequence of disorder onset may prove beneficial in terms of case conceptualization with respect to nutrition. Consideration of other psychiatric diagnoses such as obsessive compulsive, bipolar, and borderline personality disorders may prove beneficial for FA treatment as new data becomes available. We have proposed that identifying different phenotypes for FA as well as for EDs might improve nutrition interventions and even modify treatment models. An important question at this time remains unanswered: where does all the dieting stem from in the first place? How have sociocultural influences engrained highly palatable foods into the brain’s reward expectancy? Furthermore, can public health interventions aimed at reducing exposure to highly processed foods eventually reduce FA and subsequently reduce chronic dieting?

## 10. Conclusions

While there is disagreement regarding FA, it appears that much of the controversy pertains to the treatment (lacking data) rather than the existence of the problem (robust data). More specifically, nutrition interventions for individuals with FA and co-occurring ED characterized by high levels of dietary restraint are less clear than for individuals with FA and no history of restrictive ED. Individualized treatment might be helpful based on the existence of FA, but only after it has been determined that the FA signal represents an addiction to food (true positive), rather than a consequence of dietary restraint, food insecurity or insufficiency, or other forms of deprivation or food-related neglect (false positive). Dismissing FA as a clinical entity is ill informed and not helpful. FA may warrant consideration as a distinct category in the DSM, which might lead to additional research at the individual and group level, as well as public health efforts to improve the national food environment. The impact of contemporary Westernized foods may be contributing to poor ED treatment outcomes. Patients may become increasingly distrustful of the message that “there are no bad foods.” EDs are far more heterogenous than the transdiagnostic theory originally proposed, and recent data on FA supports this conclusion. Treatment models must be trauma informed. Food philosophies must be dynamic, continually incorporating new findings. In summary, one size will not fit all in FA treatment, and collaboration with the patient is crucial to develop a mutually agreeable/achievable plan. Steps and assessment tools offered herein may improve the clinical utility of FA and, in doing so, improve quality of life in individuals seeking care.

## Figures and Tables

**Table 1 nutrients-12-02937-t001:** Eight Step Process for Clinicians to Discern Food Addiction from Dietary Restraint in Order to Inform Inclusive vs. Exclusive Nutrition Strategies.

Step	Assessment	If Negative	If Positive
**1**	Food Addiction (FA) YFAS 2.0 [2] or mYFAS [204]	FA is unlikely to be a relevant construct	⇒ Step 2
**2**	Dietary Restraint Examine history of dieting behavior and role of body image as well as internalized weight bias Can use EDE-Q (long or short) [205] or EAT-26 [206] or similar validated tools	FA is likely to be a relevant construct Consider ruling out food insecurity since it is also a form of deprivation that may increase FA symptoms [207,208]	Consider if the FA preceded the restraint, or if the restraint created the FA If FA came first, it is likely to be informative that FA is a relevant construct ⇒ Step 3, and ⇒ Step 7
**3**	Substance Use Disorder (SUD) Can use clinical diagnosis or self-report or validated measure Can assess reward dysfunction by also considering addictions to caffeine and nicotine Can also assess impulsivity using BIS-11 [209,210] which may help to better understand loss-of-control behavior	Absence of other addictions does not rule out FA. However, concurrent low levels of impulsivity may suggest that the individual is unlikely to have an actual FA. Will want to also consider ADHD when assessing impulsivity ⇒ Step 4, and ⇒ Step 8	FA is likely to be a relevant construct. It is worth considering if the FA or SUD came first If SUD came first, it may indicate that inclusive nutrition strategies are the most practical ⇒ Step 4
**4**	PTSD including complex PTSD Can use clinical diagnosis or validated measure such as PCL-5 [211] Qualified professionals are required to assess the presence of CPTSD because it can be difficult for some patients to “connect the dots” across multiple life events	If there is an absence of SUD and PTSD, the presence of dietary restraint suggests that FA symptoms are driven by restriction rather than an actual FA. An exception would be if it was clear that FA preceded the restraint; however, in the absence of SUD and PTSD, inclusive nutritional strategies are likely to be the most practical ⇒ Step 6	FA is likely to be a relevant construct regardless of whether there is SUD history. However, history of SUD likely strengthens the confidence in the FA signal ⇒ Step 5
**5**	Early Life Adversity (ELA) Can use validated measures such as ACE [112], CTQ [114,115], ETI-SR [212]	Suggests an absence of biological embedding. While later life traumatic experiences can alter physiology, an absence of ELA indicates that inclusive nutritional strategies may be more plausible. There may be some cases of ELA in the absence of PTSD which can indicate high levels of biological resilience, also warranting inclusive nutritional strategies ⇒ Step 6	FA is very likely to be a relevant construct, and in the presence of ELA, PTSD, and SUD and no evidence of dietary restraint as a predisposing risk factor, exclusive/restricted nutritional strategies may be warranted, assuming there are adequate resources including social support and access to nutritious unprocessed foods
**6**	Depression Can use clinical diagnosis or self-report or validated measures such as PHQ-9 [213], BDI [214], or CESD [215]	With low levels of depressive symptoms, an inclusive nutritional strategy is likely to be the most practical strategy ⇒ Step 7	If depressive symptoms persist, it may be worth making drastic dietary changes such as the exclusion of highly processed foods in order to improve mood
**7**	Anxiety Can use clinical diagnosis or self-report or validated measures such as the BAI [216], STAI [217], GAD-7 [218] or similar validated tools	Low levels of anxiety indicate that an inclusive nutritional strategy is likely to be most practical ⇒ Step 8	Consider if anxiety is related to body image disturbance. If body image drives anxiety (or vice versa), it may indicate dietary restraint, suggesting an inclusive nutritional strategy. If anxiety is not associated with body image, improving nutritional status by excluding certain foods may be warranted (and safe)
**8**	ADHD Can use clinical diagnosis or validated measures such as ASRS [219]	If ADHD is negative but there are high levels of impulsivity, it may indicate higher likelihood of FA	Consider if eating behavior has been altered by the impact of stimulant medications

**Legend:** YFAS: Yale Food Addiction Scale; FA: Food Addiction; EDE-Q: Eating Disorder Examination Questionnaire; EAT-26: Eating Attitudes Test-26; SUD: Substance Use Disorder; BIS-11: Barratt Impulsiveness Scale-11; ADHD: Attention Deficit Hyperactivity Disorder; PTSD: Post Traumatic Stress Disorder; PCL-5: PTSD Checklist for DSM-5; CPTSD: Complex Post Traumatic Stress Disorder; ELA: Early Life Adversity; ACE: Adverse Childhood Experience; CTQ: Childhood Trauma Questionnaire; ETI-SR: Early Trauma Inventory Self-Report; PHQ-9: Patient Health Questionnaire-9; BDI: Beck Depression Inventory; CESD: Center for Epidemiological Studies Depression; BAI: Beck Anxiety Inventory; STAI: State Trait Anxiety Inventory; GAD-7: General Anxiety Disorder-7; ASRS: Adult ADHD Self-Report Scale.

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
