# Peer review of "Separating the Signal from the Noise: How Psychiatric Diagnoses Can Help Discern Food Addiction from Dietary Restraint"

_nutrients, 2020, doi:10.3390/nu12102937_

Round 1

Reviewer 1 Report

Avoiding Use of Pronouns and Editorial “We” (Example: Lines 26, 28 118, 119)

Try to avoid use of pronouns in APA writing style as a general rule, though in some instances it is unavoidable. However, in this context, I would encourage using other a specific noun or taking it out altogether (see 4.17 of APA 7th ed. )

Defining Terms (Example Lines 45 and 47)

Instead of using the (e.g. trauma) and early life adversity, I would go more in-depth by adding the actual definition of what this can include so it is less ambiguous to the reader and provides a standardized definition.

Avoiding Beginning a Sentence with Coordinating Conjunctions (Example: Line 79)

Using the mnemonic FANBOYS, most grammar critics suggest avoiding the use of For, And, Nor, But, Or, Yet, and So to begin a sentence.

Academic Wording (Example: Line 58,  95, 103)

In journal articles, authors typically want to explain abstract or broad constructs, but not “insult the readers intelligence” (though we know that this is not purposeful) by telling them we are rephrasing. Rather, I would encourage you to rephrase this or take it out altogether with phrases such as “in other words” or “clearly.”

Assumption vs. Fact (Example: Line 112-113)

Since this is a very potent statement regarding the climate of the medical/healthcare industry, I would cite a source to back up the claim.

Avoid Use of Figurative Language (Example: Line 120, 710)

Any figurative language, such as “on one hand” is discouraged, particularly for international readers or those with a different cultural background that may not fully understand the reference. For example, the use of “arm” I am unfamiliar with in the “Future Directions” portion.

Repetitive Language (Example: Line 115, 119, 122)

I typically recommend to writers that when they do a read through, that they attempt to catch repetitive phrases or words and replace them with something different by use of a thesaurus.

Nonuse of Contractions (Example: Line 156)

Avoid contractions and spell out each word individually to demonstrate scientific writing.

Numbers (Example: Line 251)

APA Style generally uses words for numbers below 10 and numerals for numbers 10 and above.

Excess Use of Quotations

There are several sections that have certain words in quotes that I believe could be taken out for clarity (ex: line 283 “go to”). With the number of abbreviations and references, it adds a wordiness that could be eliminated.

Editor Questions and Comments:

  1. I am not understanding the information provided in the abstract in line 19, “others have shown that food addiction can reduce stigma associated with obesity.” This seems counterintuitive. Does this mean that if a medical term of “food addiction” is used that it reduces the stigma of obesity?
  2. I am a bit confused on the reference to the 2003 Fairburn trans-diagnostic theory (Line 98), as the research and title suggest that diagnosis is pivotal in the comprehensive treatment of each phenotype. Is that entire paragraph something that you, as the authors, are positing is not a best-practice?
  3. In your “Future Directions” section, you may want to explore with the reader additional diagnoses that may be added to the DSM (this is merely a suggestions), such as orthorexia (which presents much like “Phenotype B”).
  4. In line 46, it is stated that the current review employs a biopsychosocial approach. However, the majority of the text is dedicated to biology and psychology (thoroughly and very well done), but has little social perspective. You may want to consider adding in a bit more that is related to the social context. Some areas that you might do this are answering the question that is posed in line 725 is where does this derive from? Mass media, the addictive nature of salt/sugar/fat on the reward mechanism of the brain which is used in most celebrations (such as cake for birthdays or pizza for parties) being engrained in the culture. Also, in the clinical vignettes, perhaps Alma being a “foodie” is more of a cultural complement to her being from Columbia, where Colombian desserts are used in various festivals and celebrations.

Reviewer 2 Report

Review of Nutrients 932020

The study discusses evidence concerning the interrelationships between food addiction, eating disorders, PTSD, and various traits, behaviors, and other psychiatric conditions. Beginning with a review of relevant evidence, and arguing for a broad and multifaceted view of eating-related clinical presentations, the study then uses three clinical vignettes to discuss clinical assessment and treatment challenges, including an eight-step process to separate the signal from the noise. The manuscript is ambitious, and the one-size-does-not-fit-all perspective is laudable, since organizing research evidence into clinical recommendations that point to heterogeneity and individualizing care is something the field sorely needs, to replace the hegemony of diagnostic classification that may hide as much as they reveal. Starting at the clinical vignettes, the manuscript takes off and becomes very readable and speaks of deep clinical experience and a breadth of perspective that is highly rewarding. The preceding theoretical overview section however, while impressive in its scope and ambition, does need work to become comparably informative and readable. I offer a few suggestions below, more or less in the order they appear in the text.  

Line 51, clarify that ”the measure itself” is the YFAS – now the phrase seems to refer to FA which is not a measure.

Sentence on lines 59-60 lacks context here, and is followed by a sentence that seems to presuppose that FA is a defined, treatable condition. The definitional issues should be handled clearly with a statement of intended use of the FA concept in the present paper. At this point in the text, the reader is unclear as to whether the authors think FA exists. In fact, it would make sense to me to place that discussion before prevalence rate studies, since before we’re clear on whether something exists and what it is, prevalence is sort of an empty term. As one reads on, it is unclear whether the authors consider FA and ED, or an addiction, or neither/both. It does seem later on that the authors use FA as a treatment-worthy clinical condition that needs to be subdivided into separate phenotypes (for example the sentence on lines 293-295), and grounding that discussion early on to counter the objection that FA isn’t a real thing is necessary for the text to work. So some kind of introduction of this issue might be worthwhile, whether to state the authors’ position and say that they will argue for it, or to say that these are questions that are yet to be answered. This would help the reader with the slight confusion that starts to grow after a few pages.

Section 1.1 is a bit disorganized and hard to follow: and feels like statements piled on top of each other, for example the stigma issue is interspersed with treatment evidence, followed by public policy. The section could be ordered into a narrative with more easily followed structure.

Line 82, missing “the” between proposed and terms. There are several places with missing words and slightly clunky language, suggest careful proofreading for better flow. I note a few language issues below but not all; proofreading is needed.

Line 83, suggest replacing “the substance-related disorder” with “FA as a substance-related disorder”

Line 91, ref 32 should have a page number for the direct quote.

Line 93, sentence should be “A study based on a large prospective cohort…”.

Lines 94-95, “In other words” seems misplaced; the obesity study does not necessarily suggest anything about why ED treatments have the focus they do.

Lines 100-103, when citing a study, use past tense “correlated” and state that a particular study showed this so as not to suggest that this is true of all women forever. Also, the sentence should start with “however” or something since it diverges from the Fairburn claim.

Line 103, “Clearly” suggests that the foregoing cited studies lead to this conclusion but they don’t; Fairburn says the opposite and ref 36 talks about weight suppression which is not the same as dieting. Maybe say “Thus, dieting may not necessarily…”.

Line 115, suggest not using ref 42, it was written by the shareholders of a for-profit treatment company who have a vested interest in claiming that all treatments except theirs are inadequate (and often uncritically do claim that). There are more neutral references for statements like this.

Line 130, is it possible to find a better term than “diagnoses” for FA, since this will make readers who know that this is not a recognized diagnosis jump. Or clearly stipulate at the outset that the term “diagnosis” is used loosely in this context (a different word is better though).

Lines 130-133 seem to suggest that people with bulimia nervosa are underweight, which they are not, in fact a good proportion are overweight or obese, and then the section discusses anorexia nervosa whereas the section heading is bulimia nervosa. It just gets confusing, I suggest the authors change the heading to include AN, and reorganize the text.

Lines 141-142, note that the study is about BN patients, and please spell out BP as binge-purge; having introduced the initialism as an AN subtype doesn’t translate smoothly to using it as a stand-alone.

Lines 142-144, the sentence needs to be anchored to its surroundings: are negative urgency, reward dependence, and lack of premeditation implicated also in FA? Otherwise, what is the relevance of this study to the overlap/interaction argument between ED and FA put forward here?

Line 154 “associates” should be “is associated”

Line 160-170, sentences are piled again, there need to be things like “In addition”, “Relatedly”, “Also”, and an argument narrative that clarifies how this logic is built. As it is, the reader has to do this and it stops smooth reading in its tracks, and is difficult to follow.

Line 168 “would focus on” should be “require”

Lines 179-180, is there a reference for this sentence, or a line of reasoning that makes it likely to true?

Sentence on 203-206, note should be made of the genetic correlations in the study and clarify the argument. Simply noting that variance in sugar consumption is explained by genetics does not automatically implicate addiction: the connection must be explicated.

Lines 217-219, this sentence may be seen as provocative and is a bit unclear: what nutrition message, and how does it hinder using insights into control system dysregulation? Also, many would suggest that “mainstream” ED treatments are very much indeed about control… This topic needs to be treated more carefully.

Line 227 “emerges” – again, use past tense language when describing previous studies.

Line 233, “it is likely” is too strong: mediation of obesity from impulsivity by FA does not make a strong statement about the causes of ED.

Line 237 and forward, piling statements again. What exactly is the relevance of impulsivity being associated with PTSD and addiction, for the relationship between addiction, FA, ED and impulsivity? Likewise, the relevance of ref 99 a few lines before is unclear. These studies are inserted without explaining why, and the line of argument develops a severe limp. Also in this section, explain what 5HTTLPR and ELA are (everyone has forgotten ELA at this point in the text, I suggest introducing the initialism here instead of on line 47).

Lines 253-254, what is the relevance of annual costs for the present argument? Also lines 265-267 about abuse suggesting more abuse; the relevance for the line of thought is unclear and it contributes to the limp again.

Lines 254-256, is ACE a questionnaire? It is presented more as a concept, but here it sounds like a measure. Clarify please.

Line 269, food is not in itself an addiction. More stringent language would help. Also the last sentence here on 269-270 is grammatically a bit off and it is not clear what it means or how it is relevant.

Line 272 and forward, the beginning of the Addictions section is a bit unclear. How are DA turnover, amphetamine response and FA mediating BMI in response to stress related? Also, the statement on lines 279-280 is not really shown and therefore not “clear” from the foregoing, as well as the following sentence saying that alcohol is not one of the addictions that biological embedding of ELA can lead to. The text overall makes sweeping statements/conclusions from disjointed snippets of evidence, and needs a lot more glue to bring the reader along.

Sentence on 313-314 needs a reference or two.

Lines 333-334, strictly speaking, no evidence that poor diets lead to depressive symptoms was presented in the foregoing paragraph, only associations that are not necessarily causal (eg trait emotion dysregulation may lead to both poor diets and depression). If there were covariates or other factors in the cited studies that strengthen a causal interpretation, these should be presented. Otherwise, the authors should be explicit that they are speculating.

Line 371, treatment for what? FA?

Section 5.3, ref 183 seems to argue against the general line of reasoning in this section, or at least the implied interpretation of other cited studies. Some comment regarding direction of effect and its possible relevance might be warranted.

Table 1 – the “2” by “If Positive” cell is unclear. When should the practitioner move to step 3 and when to step 7? Or should both be investigated? Overall, the bullet points are unclear and other symbols or language to help direct the flow of activities would be helpful.

Round 2

Reviewer 2 Report

The authors done a careful and thorough job in addressing my concerns.

Author Response

Please makes these edits prior to publication:

Please change to person first language (individuals/patients with obesity/ED/FA) in the following lines: 
42, 116, 226, 233, 241, 312, 393, 395, 733 
We have corrected all of these and one other (line 237).

Line 36 – change Statistics to “Statistical” 

Done.

Line 147 – An intervention study among 148 women with BN (n=66), those with higher FA severity at baseline were less likely to obtain abstinence 149 from binge-purge episodes following treatment – add “In” at the beginning of the sentence or "reported that" before those

We chose the first recommendation (in).

Line 192 - engage dysfunctional food-related behaviors – add “in” after engage

Done.

Line 431 - never engaged any compensatory behaviors – add “in” after engaged

Done.

Line 459 - diagnosed ADHD – add “with” after diagnosed

Done.

Line 464 – private school high school – change to “private high school”

Done.

Line 481 - but as be became increasingly concerned about his appearance – change be to “he”
Done.

Line 515 - Whitney begun to express suicidal ideation – change begun to “began”
Done.

This manuscript fits very well within the scope of this special issue on Clinical Utility of Food Addiction, and will be of great interest to readers.  

Thank you for the thorough and excellent review.
